# DFMTDS: Distribution-free martingale test Distribution shift

## Abstract

A standard assumption in machine learning is the data are generated by a fixed but unknown probability, which is equivalent to assuming the examples are generated from the same probability distribution independently. So for most of the learning research we usually randomly shuffle the whole data into training and test data set. However, for real-life application reality is that the data points are observed one by one. This paper is devoted to testing the assumption of distribution shift on-line: the observed data arrive one by one, and after receiving each object, the machine learning algorithms give a prediction label, we would like to have a valid measure of the degree to which the evidence to against the assumption of non-distribution-shift. Such measures are provided under the framework of distribution-free methods, also called martingales measure, which is a general empirical theory of probability developed in 1993-2003. We report the experimental performance of martingales to measure on the real-life data sets and the results to show a bona fide fact that the distribution shift testing is an inescapable reality when we adaptive machine learning algorithms to the original order.

## 1 Introduction

The rapid development of learning theory technology recently has made it possible to solve ever more difficult problems in real-life problems. The development of high-performance computing algorithms has been essential to this progress. Generally speaking, the problem of learning can be reduced into the problem that estimates the dependence based on empirical data, which is practically complete by means of machine learning algorithms as depending on the assumption that the data are generated independently identify distribution (i.i.d). In order to state and perform state-of-the-art properties of such algorithms, it is the almost standard announcement that the data satisfy the i.i.d. assumption. We are concerned with the next natural step is that, however, the real demand of application must make the sense that the data are generated one by one. So it is surprising that so little work has been done on testing the distribution shift when we adaptive the learning algorithms to make predict tasks.

The problem of distribution shift detection has usually a significant impact on the learning theory research community. In on-line setting, observed data are obtained one by one from a sensor source and the assumption of i.i.d of the data means that data are generated by a fixed but unknown distribution in the data collection process. In the statistics research community the problem is generally known as the *change-point detection* problems. The challenge of change-point detection, under the shackles of traditional statistics, is that classical statistics methods could never overcome the curse of dimensionality. However, as proposed in this paper, the distribution-free martingale framework approach could.

To address this challenge we are eager to assimilate the idea of *a general empirical theory of probability*, introduce by Vovk (1993) and extend the idea of testing exchangeability on-line to a martingale framework in Vovk et al. (2003). Generally speaking, a way of on-line distribution shift testing is by employing the theory of conformal prediction (Gammerman & Vovk, 2006) and then calculating exchangeability martingales against the assumption. For more details that there are two steps to achieve the goals, that the first step is to implement a transductive conformal predictor, which outputs can be used for constructing a sequence of $p$-values functions. Secondly, we introduce exchangeability martingales, which are the measure of $p$-values function, and track the deviation

between the assumption and the empirical performance. Once the martingale grows up to a pre-defined large value (20 and 100 are convenient rules of the level), we could have sufficiently large evidence against the assumption of fixed distribution, or in other words that we have sufficiently large evidence to confirm the distribution shift phenomenon.

In this paper we adaptive the exchangeability test to distribution shift detection. The first procedure of testing exchangeability on-line is described by Vovk et al. (2003) for the USPS data set. The core motivation is based on a new general empirical theory of probability, which was proposed by Vovk (1993). Vovk (2020; 2021) adapt conformal e-prediction to change detection based on techniques of conformal prediction. Ho (2005) applies power martingales to the problem of change detection in time-varying data streams. Ho (2012) describes the detection of concept changes by testing exchangeability. Ho et al. (2019) proposes a martingale-based approach for flight behavior anomaly detection. The author shows that the martingale detection approach is an efficient and one-pass incremental algorithm and can be useful to deal with real-life high dimensional data.

We propose a framework to test the distribution shift for any transducers. As following that for the first step, select transducers, secondly, extend conformal prediction methods to compute $p$-value for each example, and finally choose the significance threshold $\varepsilon$ for betting function, which determines the contribution of $p$-values to the value of the martingale.

The rest of the paper is organized as follows. In Section 2 we review the concept of martingale and give a definition of distribution-shift martingales. Section 3 presents the construction of power martingales, which does not dependent on the specified distribution. Section 4 shows experimental results of testing three real-life data sets for distribution shift detection; We compare four algorithms, such as Support Vector Machine, Boosting, and Neural Network, under two situations, as standard randomly permuted and the original one-by-one order. The conclusion show in Section 5.

## 2 DISTRIBUTION-FREE MARTINGALES

This section outlines the necessary definition and concepts of martingales.

### 2.1 DISTRIBUTION-SHIFT

Consider a sequence of random variables $(Z_1, Z_2, \ldots)$, the corresponding outputs elements $(z_1, z_2, \ldots)$ called *examples*. It is often the case that each example consists of two parts: an *object* $x_i$, and its *label* $y_i$. The hypothesis of *Distribution-shift* is that the examples $z_1 z_2 \ldots$ are generated by a *fixed but unknown* probability distribution $Q$.

### 2.2 MARTINGALES FOR DISTRIBUTION-SHIFT TESTING

We are interested in testing the hypothesis of distribution-shift *on-line*: after observing each new example $z_n$, the transducer is required to output a number $M_n$ reflecting the strength of evidence against the hypothesis. The support we want to test the hypothesis that the data is generated by a fixed but unknown distribution dependently, that is equivalent to the *null* hypothesis is that:

$H_0$: *There is no distribution-shift.*

When we test the hypothesis we would construct a *supermartingale*, which is a sequence of random variables $\{M_0, M_1, \ldots\}$ and

$$M_n \geq E(M_{n+1}|\mathcal{F}_n), \tag{1}$$

where $\mathcal{F}_n$ is the $\sigma$-algebra generated by $z_1, \ldots, z_n$. If $M_0 = 1$ and $\inf_n M_n \geq 0$, as enlightened and proposed by Vovk (1993; 2001; 2021), Bienvenu & Shen (2009), and Bienvenu et al. (2009), $M_n$ can be regarded as the *capital process* of a player who starts from 1, that the player convinces that he will never risks bankruptcy, at the beginning of each trial $n$ places a fair bet on the $z_n$ to be chose by Nature, which equivalent to the implement of learning algorithms. If such a supermartingale $M_n$ ever takes a large value, our belief in $Q$ is undermined and we could have rejected the null hypothesis, which intuition is formalized by Doob's inequality,

$$Q\{\forall n : M_n \geq C\} \leq \frac{1}{C}, \tag{2}$$

where $C$ is an arbitrary positive constant. The proof about it can be found at Doob (1984).

From the viewpoint of supermartingale, it is unlikely for any $M_n$ to have a large value. For the problem of test distribution shift, if the final value of collective data of a martingale is large then the null hypothesis can be rejected with the corresponding valid probability.

## 2.3 CALCULATION $p$-VALUES

Let $(z_1, z_2, \ldots)$ denote a sequence of examples arrived as the original order. The calculation of $p$-values for the given examples can adopt the conformal prediction methods developed by Vovk et al. (2005). Following the general idea of transductive inference (Vapnik, 1995; 1998), that the empirical facts can be *transductive* into the values of a function. Conformal prediction, informally naming *Transductive Confidence Machine* (Vovk et al., 2003), can be adopted for the top of any transducers to generate a sequence of $p$-values corresponds to examples. A measurable function, namely *nonconformity scores* defined,

$$\alpha_i = A_n(\{z_1, \ldots, z_{i-1}, z_{i+1}, \ldots, z_n\}, z_i). \tag{3}$$

In this paper, four different algorithms are implements, such as SVM, Boosting, and the multilayer perceptron neural network (MLP), to compute the nonconformity scores. There are many different ways of defining nonconformity measures, and a natural measure of nonconformity of $z_i$ are adopt the *inverse probability* of prediction. The *p-value* definition for $z_n$,

$$p_n = \frac{\#\{i : \alpha_i > \alpha_n\} + \tau_n \#\{i : \alpha_i = \alpha_n\}}{n},$$

where $\tau_n$ is uniform distribution values from [0,1] and the symbol # means a measure of set size. It is clear that for the any transducers, the $p$-values are exactly valid. The proof of Theory 1 established in Vovk et al. (2005).

**Theorem 1.** *If examples $(z_1, z_2, \ldots)$ satisfy exchangeability assumption, the sequence of p-values $(p_1, p_2, \ldots)$ that are independent and uniformly distributed in $[0, 1]$.*

The property that the example generated a sequence of $p$-values by conformal prediction, is exactly valid uniformly distribution, which allows us to make statistical inference.

## 3 DISTRIBUTION-FREE MARTINGALES BASED ON $p$-VALUES

This section focuses on the second step of testing: given the sequence of $p$-values, we can test distribution shift by calculating distribution-free martingales as the function of the $p$-values.

For each $i \in \{1, 2, \ldots\}$, let $f_i : [0, 1]^i \to [0, \infty)$ be a *betting function*,

$$\forall i : f(p_i) = \varepsilon p_i^{\varepsilon-1}, \tag{4}$$

where $\varepsilon \in [0, 1]$. Since $\int_0^1 \varepsilon p^{\varepsilon-1} dp = 1$, the random variables

$$M_n^{(\varepsilon)} := \prod_{i=1}^{n} (\varepsilon p_i^{\varepsilon-1}), \tag{5}$$

where $p_1, p_2, \ldots, p_n$ are the sequence of $p$-values calculated by conformal predictors, will be a non-negative randomized exchangeability martingale with initial value 1; this family of martingales, indexed by $\varepsilon \in [0, 1]$, will be called the *randomised power martingales*, which does not dependent on the specified distribution. This method is named *Distribution-free martingale test Distribution shift (DFMTDS)*. Algorithm 1 summaries the process of distribution shift testing on-line.

## 4 EMPIRICAL STUDIES

In this section, we investigate the performance of distribution shift detection in with that of the distribution martingale. Three real-life data sets have been tested under three transducers, such as SVM, neural network, and Boosting respectively: the `USPS` data set, the `Satimage` data set, and the `Segment` data set.

---

**Algorithm 1** DFMTDS: Distribution free martingale test Distribution shift

---

**Input**: $(z_1, \ldots, z_l)$
**Output**: $M_n$
 1: Let $i \in \{1, \ldots, n\}$:.
 2: compute $\alpha_i$,
 3: compute $p_i$,
 4: compute $M_i$,
 5: **return** $M_n$

---

Table 1: Main dimensional characteristics of data sets

| Data Set | # of Examples | # of Attributions | # of Classes |
|----------|---------------|-------------------|--------------|
| USPS     | 9298          | 256               | 10           |
| Satimage | 6430          | 36                | 6            |
| Segment  | 2310          | 19                | 7            |

`USPS` is a handwritten digits from 0 to 9, of which were collected from real-life zip codes. Each example is described by $16 \times 16$ grey-scale images. This data set includes 9298 examples, of which 7291 examples are marked as the training set and 2007 examples are marked as the testing set. Each example contains 256 grey-scale values of all pixels as features. All the examples are labeled from "0" to "9".

It is well-known that the USPS data set is heterogeneous; in particular, the training and test sets seem to have different distributions (Vovk et al., 2003). In the next subsection, we will see the huge scale of the martingale of this heterogeneity.

`Satimage` is the Landsat Satellite Image data set provided by Ashwin Srinivasan, University of Strathclyde. Each example consists of 36 attributes of $3 \times 3$ pixels in the neighborhood from 4 multi-spectral scanner images as empirical observations. This data set includes 6430 examples, of which 5042 examples are marked as the training set and 1388 examples are marked as the testing set. The whole data set is labeled into 6 classes. `Satimage` is given in random order and certain lines of data have been removed, so it is established exactly that we will obtain the same scale of the martingale if we randomly shuffle the Satimage data set.

`Segment` is the Image Segmentation data set by Vision Group, University of Massachusetts. The examples were *drawn randomly* from a database and hand-segmented into $3 \times 3$-pixel pieces. This data set includes 2310 examples, of which 1811 examples are marked as the training set and 499 examples are marked as the testing set. The whole data set is labeled into 7 classes. `Segment` has been drawn randomly from a database, so the performance of martingale will establish the same scale after randomly shuffling.

Some dimensional characteristics of these data sets including the size of data sets, the number of attributes, and the number of classes are summarised in Table 1, the data set split information is listed in Table 2.

## 4.1 THE SELECTION OF $\varepsilon$

When we applied to the selected transducers, such as SVM, MLP, and Boosting, the family of distribution-free randomized power martingales might at first not look very promising (Figure 1

Table 2: Main training and test split of data sets

| Data Set | # of Examples | # of Training | # of Test |
|----------|---------------|---------------|-----------|
| USPS     | 9298          | 7291          | 2007      |
| Satimage | 6430          | 5042          | 1388      |
| Segment  | 2310          | 1811          | 499       |

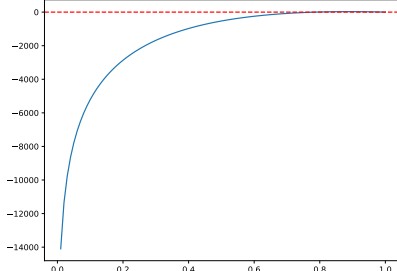 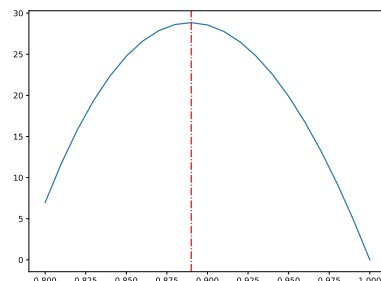

Figure 1: The final values, on the logarithmic (base 10) scale, attend by the randomised MLP power martingales $M_n^\epsilon$ on the USPS data set. *Left* the full parameter $\epsilon$, and *right* the zoom-in range of the parameter $\epsilon$

Table 3: Significant threshold selection for on-line learning

| Data Set | SVM | MLP | Boosting |
|---|---|---|---|
| USPS | .77 | .89 | .94 |
| Satimage | .96 | .92 | .91 |
| Segment | .98 | .99 | .95 |

[left]), but if we concentrate on a narrow range of $\epsilon$ (Figure 1 [right]), it becomes clear that the final values for some $\epsilon$ are very large. In this paper, we use the martingales to monitor the trajectory of distribution, so we select the significance values of $\epsilon$. The significant threshold selection for on-line learning and randomly shuffling protocol are list in Table 3 and Table 4 respectively.

## 4.2 EMPIRICAL RESULTS

Figure 2 shows the martingale trajectory performance of the randomized transducers power martingale. The left panel is the full USPS data set adaptive the SVM, Boosting, and MLP as the transducer respectively. The right panel is an empirical performance for randomly shuffling protocol.

Figure 2 left panel shows the trajectory performance of the martingales when the examples arrive in the original order: first the 7291 of the training set and then 2007 of the test set. The $p$-values are generated on-line by the DFMTDS algorithm and three transducers power martingale, such as SVM power martingale, MLP power martingale, and Boosting power martingale are calculated respectively. The final value of the SVM power martingale is $\approx 10^{150}$, the final value of MLP power martingale is $\approx 10^{48}$, and the final value of Boosting power martingale is $\approx 10^{52}$. It is equivalent to explaining from Doob's inequality that we have strong evidence against the null hypothesis.

Figure 2 right panel shows the trajectory performance of the martingales when the examples are randomly shuffling. As excepted, the final value of the SVM power martingale is $\approx 10^{62}$, the final value of MLP power martingale is $\approx 10^{18}$, and the final value of Boosting power martingale is $\approx 10^{19}$. Empirically, once the final value grows up not to a large value (20 and 100 are convenient rules of thumb) (Fedorova et al., 2012), In simple words that the martingales do not reject the null hypothesis.

Table 4: Significant threshold selection for randomly shuffling learning

| Data Set | SVM | MLP | Boosting |
|---|---|---|---|
| USPS | .84 | .92 | .91 |
| Satimage | .96 | .99 | .96 |
| Segment | .98 | .99 | .95 |

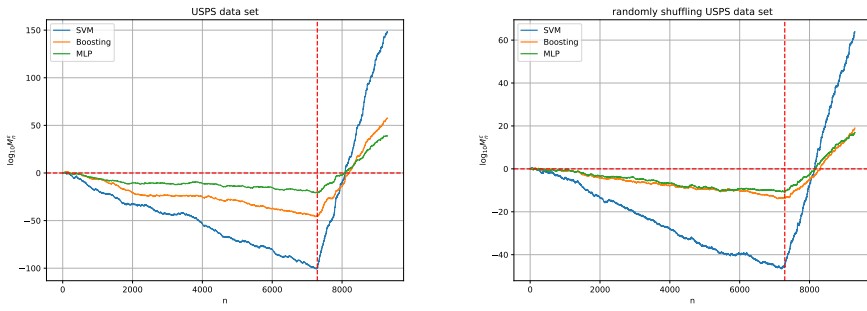

Figure 2: `USPS` data set performance

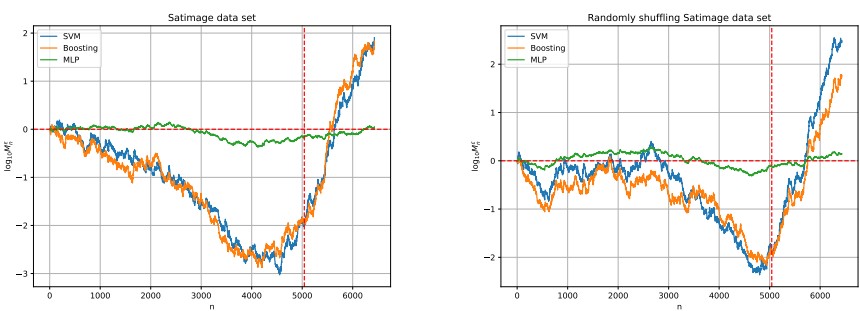

Figure 3: `Satimage` data set performance

Figure 3 shows the trajectory performance of the martingales of the Satimage data set. The largest final value for the transducer martingale is $\approx 10^2$ for the original order setting and $\approx 10^{1.2}$ when randomly shuffling. It is argued as expected that we have no evidence against the hypothesis for the Satimage data set.

Figure 4 shows the trajectory performance of the martingales of the Segment data set, As expected that we have no evidence against the hypothesis for the Segment data set.

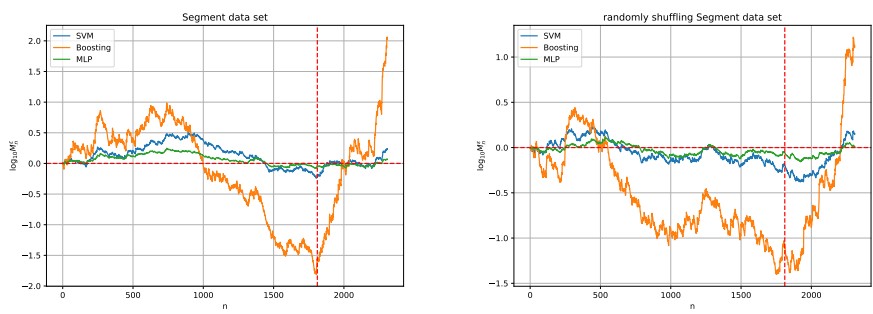

Figure 4: `Segment` data set performance

## 5 DISCUSSION AND CONCLUSIONS

This paper introduces a new way of martingales framework for distribution shift test by means of distribution-free methods. We have shown that propose methods can be adapted to any underlying algorithms as the transducers and then by means of conformal prediction we can obtain the valid $p$-values and then calculate the transductive power martingale. The martingales framework distribution-shift detection based on the betting function provides an exact solution for betting against the hypothesis.

Our goal has been to find distribution-free martingales that do not need any assumption about the data distribution and can be adapted to any high-dimensional problems. The final empirical evidence shows that for the concrete application with help of machine learning methods, martingale testing plays a bona fide role.

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
