# OpenReview forum: "DFMTDS: Distribution-free martingale test Distribution shift"
_ICLR.cc/2022/Workshop/EmeCom — Submitted to EmeCom Workshop at ICLR 2022_

### Official Review · Reviewer_eocA · 2022-03-20
**Interesting work but not related to EC**

**Rating:** Strong rejection
**Confidence:** 4

**Review:**

This work evaluates whether an online learning problem has a distribution shift using martingale measures.

Though EC does have some relation to continual learning and online learning (language drift, multitask learning, multimodality), it is not particularly close to the brand of online learning + distribution shift problem that is presented in this work. The work also makes no mention of relating to the field of EC. Furthermore, this work is 6 pages long as opposed to the requirement of 4 pages and seems completely unchanged from a similar ICLR submission.

I don't believe this would make for interesting discussion at the workshop but would be eager to hear if I have misunderstood.

---

### Official Review · Reviewer_SkW1 · 2022-03-21
**Interesting framework but not within the scope of the workshop**

**Rating:** Rejection
**Confidence:** 3

**Review:**

In this paper, the authors tackle the proper of distributional shift by introducing a new way of martingales framework leveraging distribution-free methods. Although semantic drift in communication games is a kind of distributional shift, I don't see the work and approach discussed as being directly applicable to the best of my knowledge. Nor do the authors present their work toward tackling these problems in the context of language games and emergent communication. That is why I consider the work not well aligned with the scope of the workshop.

---

### Decision · Program_Chairs · 2022-03-25

**Decision:**

Reject

**Comment:**

This paper is not relevant to the workshop and also does not follow the style guidelines